# Seismological evidence for a multifault network at the subduction interface

Caroline Chalumeau[1], Hans Agurto-Detzel[1], Andreas Rietbrock[1 ✉], Michael Frietsch[1], Onno Oncken[2], Monica Segovia[3] & Audrey Galve[4]

Subduction zones generate the largest earthquakes on Earth, yet their detailed structure, and its influence on seismic and aseismic slip, remains poorly understood. Geological studies of fossil subduction zones characterize the seismogenic interface as a 100 m–1 km thick zone[1–3] in which deformation occurs mostly on metres-thick faults[1,3–6]. Conversely, seismological studies, with their larger spatial coverage and temporal resolution but lower spatial resolution, often image the seismogenic interface as a kilometres-wide band of seismicity[7]. Thus, how and when these metre-scale structures are active at the seismic-cycle timescale, and what influence they have on deformation is not known. Here we detect these metres-thick faults with seismicity and show their influence on afterslip propagation. Using a local three-dimensional velocity model and dense observations of more than 1,500 double-difference relocated earthquakes in Ecuador, we obtain an exceptionally detailed image of seismicity, showing that earthquakes occur sometimes on a single plane and sometimes on several metres-thick simultaneously active subparallel planes within the plate interface zone. This geometrical complexity affects afterslip propagation, demonstrating the influence of fault continuity and structure on slip at the seismogenic interface. Our findings can therefore help to create more realistic models of earthquake rupture, aseismic slip and earthquake hazard in subduction zones.

On 27 March 2022, a 5.8 moment magnitude ($M_w$) earthquake occurred at a depth of 19 km in the erosive Ecuadorian subduction margin, near the coastal town of Esmeraldas. The earthquake triggered a short-lived but highly productive aftershock sequence. At the time, a dense array of 100 short-period seismometers (Fig. 1 and Extended Data Fig. 1) was in place for the continuing offshore seismic experiment 'high-resolution imaging of the subduction fault in the Pedernales Earthquake Rupture zone' (also known as HIPER).

We detect and locate more than 1,500 events in the epicentral area between 12 March and 26 May 2022 (Fig. 1) using a combination of machine learning techniques[8,9]. Earthquakes are subsequently relocated in a three-dimensional (3D) velocity model[10] with a double-difference algorithm[11], providing an image of the subduction interface seismicity at an unprecedented resolution. The seismicity clearly fell primarily along the megathrust at depths between 16 and 23 km in a 20 km wide along-strike region. Focal mechanisms show mostly oblique thrust faulting with a small right-lateral component. The earthquake locations broadly match the plate interface modelled by Font et al.[12], and are consistent with previous offshore seismic studies[13]. A secondary cluster of seismicity associated with shallow normal faulting occurred in the upper plate, to the south-west of the mainshock, and was active intermittently throughout the observation period but at different times from the main earthquake sequence at the plate interface.

Fitting a plane to earthquakes with a location uncertainty below 75 m (Methods), we find that 90% of all well-located seismicity is contained within an 870 m thick band with a strike and dip of 18° and 23°, respectively. However, seismicity is not purely planar (Fig. 2 and Extended Data Fig. 2) as a geostatistical analysis shows that topographic features have a characteristic width of 1.8 km and a characteristic height of 210 m, whereas the thickness of the seismicity is about 240 m (Methods). This is smaller than what has been previously observed seismologically in subduction zones[7], probably due to our better resolution. It is, however, comparable to geological estimates of the thickness of fossil plate interface zones at seismogenic depths[1,3,4,14]. It also matches geophysical estimates of active plate interface thickness[15], although studies imaging the deeper part of the seismogenic zone at a high resolution are rare. We infer that the observed band of seismicity occurred primarily within the plate interface region, although some seismic activity within damaged portions of the upper and lower plate cannot be excluded.

## Detachment fault versus broad deformation

The subduction interface can be viewed as a region in which the distribution of strain is heterogeneous, containing both strongly deformed zones and domains experiencing little deformation[4,16]. Within this region, it remains debated whether seismic slip concentrates on a single detachment fault, or whether it is distributed within the plate interface region[17]. Some studies have found evidence of seismic deformation concentrated along the roof fault zone, whereas aseismic slip is distributed within the subduction channel mélange[4,18]. Others have found

[1]Karlsruhe Institute of Technology, Karlsruhe, Germany. [2]GeoForschungsZentrum (GFZ), Potsdam, Germany. [3]Institute of Geophysics, Escuela Politecnica Nacional, Quito, Ecuador. [4]Université Côte d'Azur, CNRS, Observatoire de la Côte d'Azur, IRD, Géoazur, Valbonne, France. ✉e-mail: rietbrock@kit.edu

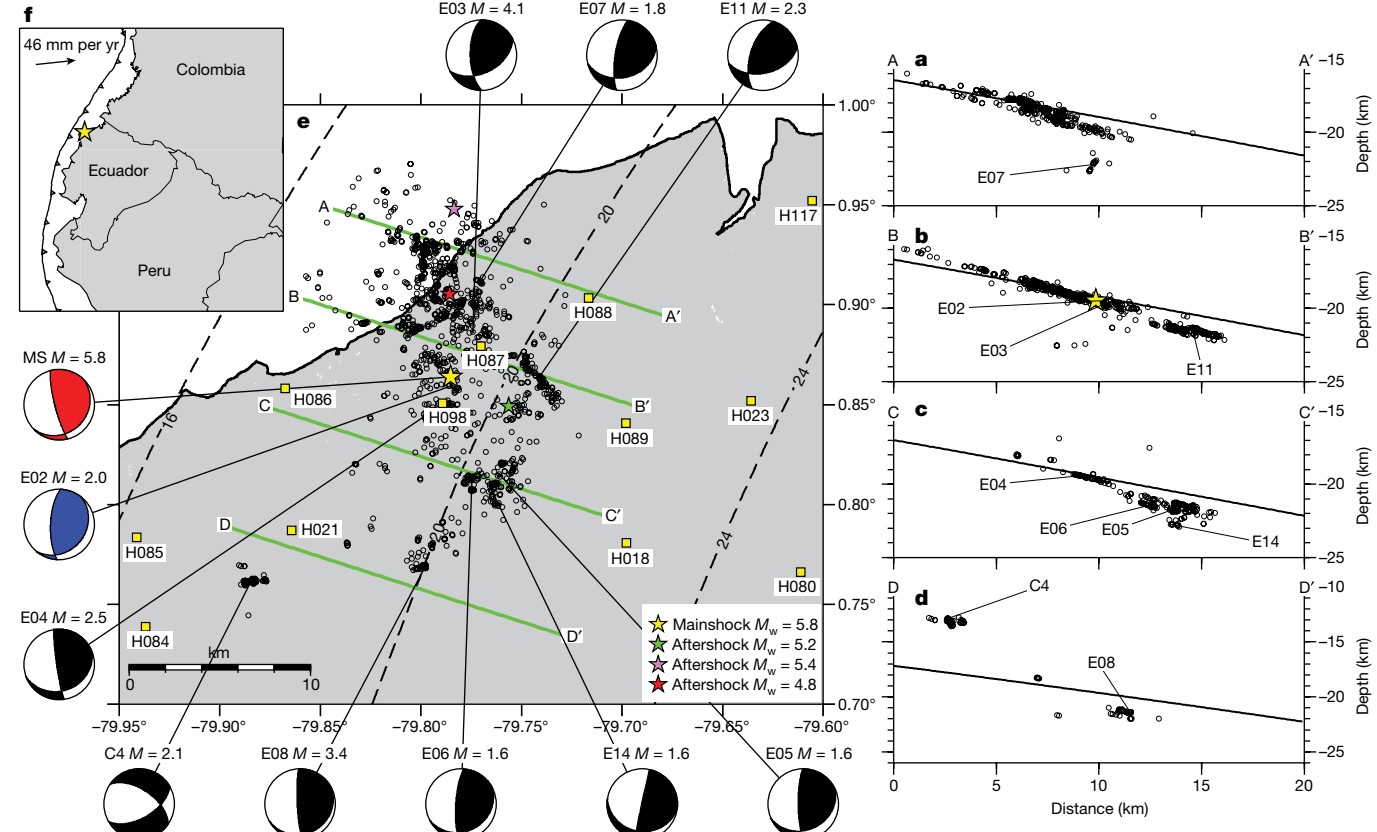

**Fig. 1 | Map and cross-sections of the Esmeraldas sequence. a**–**e**, Cross-sections (**a**–**d**) and map (**e**) of earthquakes with a location error less than or equal to 200 m, between 12 March and 26 May 2022. Cross-sections shown in **a**, **b**, **c** and **d** are marked on the map in **e** as green lines A–A′ B–B′, C–C′ and D–D′. The stars show the position of the mainshock (yellow) and the three largest aftershocks (green, pink and red). A subset of 11 well-defined focal mechanisms is shown, including the United States Geological Survey focal mechanism of the mainshock (in red) and the focal mechanism of the largest direct foreshock (in blue).

The locations for these focal mechanisms are also shown as labels in the cross-section. The slab model from ref. 12 is plotted as a black line in each cross-section and as dashed contours in map view. Nearby seismic stations are shown as yellow squares. **f**, The inset shows the convergence velocity between the Nazca plate and the North Andean Sliver. Cross-sections **a**–**d** each have a width of 5 km and a bearing of 108°, corresponding to the dip direction of the main plane of seismicity.

evidence for several anastomosing fault strands hosting seismic slip within a structural width of hundreds of metres[1,19]. It is, however, difficult to determine whether the activity on these different fault strands is concurrent[1,14], or if the different fault strands represent the gradual stepping down of the plate interface over many seismic cycles[5,6].

Our data show seismicity occurring on several distinct, often sub-parallel fault planes (Fig. 2). These planes are identified by using the three-point method on families of similar earthquakes (Methods). They are mostly between 0 and 40 m thick with a median of 21 m, which is close to our resolution limit but similar to the thickness of individual fault zones found within subduction shear zones[1,3,6,20]. We can confirm that in some places several parallel planes are active at once, whereas in others seismicity is concentrated on only one plane. The superposition occurs in two main areas: the northern area of seismicity (Fig. 2b), and downdip of the mainshock rupture (Fig. 2d). Meanwhile, the area south of the epicentre is characterized by a single, thin plane of seismicity (Fig. 2c). Variations in the thickness of the reflective band interpreted as the plate interface have been previously observed at depth by seismic imaging[15,21]. For example, in Alaska, seismic imaging shows a marked increase in thickness associated with the change from a singular localized shear zone in the seismogenic zone (100–250 m thick low-velocity zone) to a wide deformation zone downdip (roughly 4 km thick zone with several low-velocity zones)[15]. Here in Ecuador, we show that seismic slip in the seismogenic zone itself is not necessarily concentrated on a single main fault, and that the distinction between single-fault

and superposed-faults regions is not linked to depth at this scale. This complexity of the plate interface geometry probably has repercussions on the propagation of seismic and aseismic slip in the region and on the earthquake rupture process.

## Spatio-temporal evolution of seismicity

Aftershocks at the plate interface occurred around the epicentre, propagating primarily in the north-north-west direction that is neither aligned with the strike nor the dip of the megathrust (Fig. 3a). The timing of these earthquakes follows the modified Omori law, with a $P$ value of 0.99 (Extended Data Fig. 4). However, the area they covered increases with log of time (Fig. 3b), as does their cumulative number (Extended Data Fig. 5), indicating afterslip as the likely mechanism behind their occurrence[22,23]. Fluid diffusion cannot be the driver for the aftershock expansion, as the area would have to increase linearly with time[24], which we did not observe. Figure 3b also shows that the aftershock area expansion with time does not fit the model by Perfettini et al.[25] well. Instead, it shows far more complexity, slowing and speeding up and in one case perhaps overtaking the afterslip front[26]. The aftershocks first occurred near the mainshock epicentre, initially spreading mostly south but gradually expanding in other directions. After about 12–20 min, the area covered by the aftershocks reaches the expected rupture size for a $M_w = 5.8$ earthquake[27,28]. Downdip of the epicentre, a $M_w$ 5.2 aftershock occurred 85 min after the mainshock and lead to

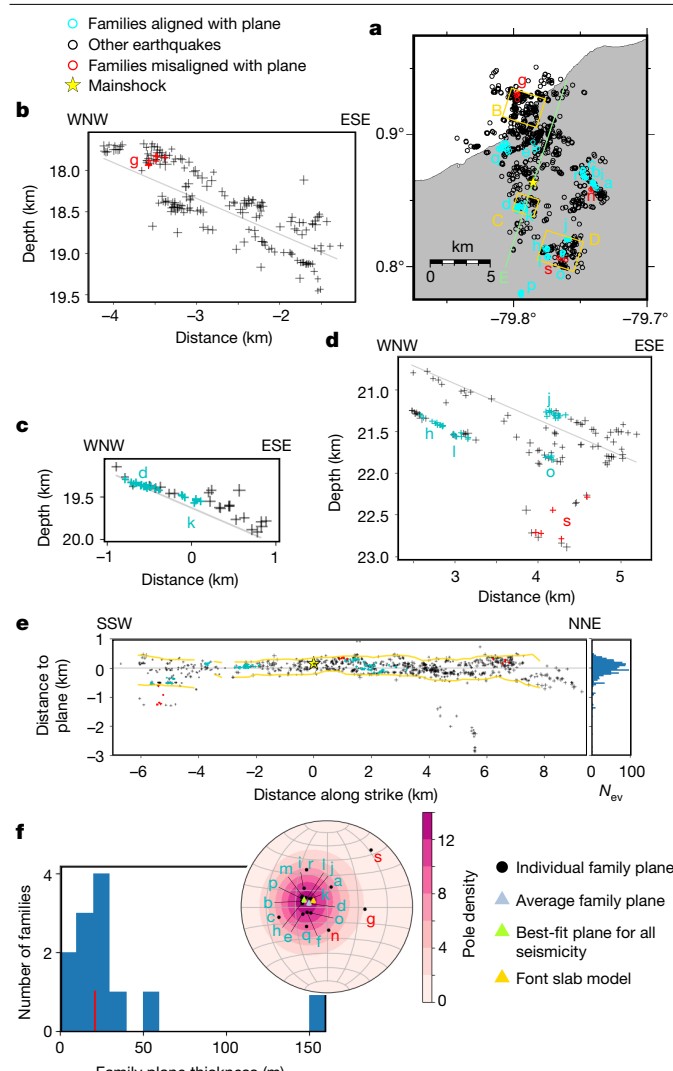

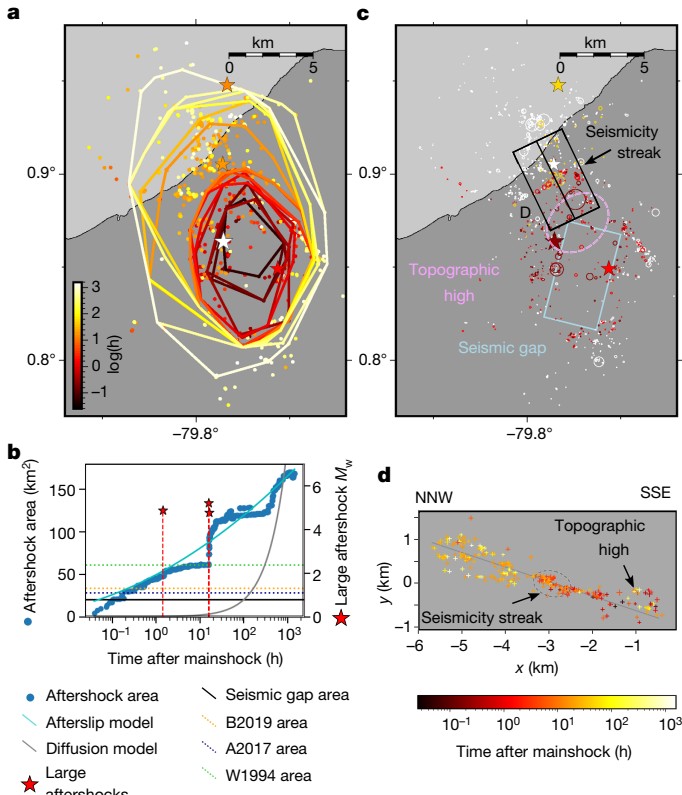

**Fig. 2 | Map and cross-sections of earthquakes with location errors less than 75 m, showing the detailed structure of the seismicity. a**, Map of all cross-sections. Cross-sections B–D all have a bearing of 108°, corresponding to the dip direction of the main plane of seismicity. Cross-section E has a bearing of 18°, corresponding to the strike direction of the main plane of seismicity. Blue and red circles show families of similar earthquakes (CC > 0.75), aligned with the main plane of seismicity with an angle lower or higher than 30°, respectively. Each family shown here is assigned a lowercase letter for identification. **b–d**, Small cross-sections of regions B–D marked in yellow in the map in **a**, all at the same scale. The size of the crosses represents the earthquakes' relative relocation error. Blue and red crosses show families aligned with the main plane of seismicity with an angle lower or higher than 30°, respectively. More cross-sections are shown in Extended Data Fig. 3. **e**, Cross-section of earthquakes along the strike of the seismicity, for the whole region. 95% of earthquakes are found between the yellow lines. Blue and red crosses show families aligned with the main plane of seismicity with an angle lower or higher than 30°, respectively. A histogram of earthquake depths relative to the plane is shown on the right, with the same vertical scale. **f**, Plots showing the thickness and orientation of the fault planes defined by families of similar earthquakes. The thickness plot only includes families with at least five events with relative location errors less than 20 m among them.

a small acceleration of the aftershock front, although seismicity did not expand far in that direction. North of the mainshock, a $M_w$ 5.4 and a $M_w$ 4.8 aftershock occurred about 16 and 16.5 h after the mainshock. Before these two earthquakes, the aftershock propagation slowed down near the location of the $M_w$ 4.8 aftershock (Fig. 3a,c). In fact, a

**Fig. 3 | Evolution of aftershock area at the interface (for events with location errors less than or equal to 200 m). a**, Map of the aftershocks and aftershock front coloured by time. Each contour is the convex hull containing 95% of events at a given time. Stars represent the mainshock (white) and the largest aftershocks (coloured by time). **b**, Aftershock area as a function of log(time) (left-hand axis). Red stars show the times and magnitudes of the largest aftershocks (right-hand axis). Horizontal lines are estimations of the mainshock rupture area from scaling relations (dashed lines) or from the estimated size of the seismic gap (full black line). The full blue line shows the best-fit model assuming afterslip drives aftershock expansion (propagation velocity of 1.1 km per decade). The full grey line shows the best-fit model assuming fluid pressure diffusion drives aftershock expansion (diffusion coefficient of 1.42 m² s⁻¹). Scaling relations: B2019, ref. 28; W1994, ref. 49; A2017, ref. 27. **c**, Aftershocks coloured by their time relative to large aftershocks. The stars show the mainshock (dark red) and the $M_w$ 5.2, $M_w$ 5.4 and $M_w$ 4.8 (light red, orange and white) earthquakes. Circles show the rupture areas of aftershocks assuming a stress drop of 3 MPa (ref. 41), occurring after the mainshock (dark red), $M_w$ 5.2 aftershock (light red), $M_w$ 5.4 aftershock (orange) and $M_w$ 4.8 aftershock (white). The light blue box represents a gap in seismicity, which serves as a first-order approximation of the mainshock rupture. **d**, Earthquakes coloured by time, projected onto a plane orthogonal to the streak indicated in **c**. The projected location of the seismicity streak is indicated as a dashed ellipse, and the projected main plane of seismicity is shown as a dark grey line. Timelapses are shown in Supplementary Videos 1 and 2.

north-east–south-west alignment of seismicity appeared during this time in this region (Fig. 3c,d). After these two aftershocks, there was a large increase in aftershock production and a notable acceleration of the aftershock expansion in the northern direction. The aftershock area eventually stopped expanding about 2.5 days after the mainshock, when it reached around 120 km². Subsequently, small (M < 4) earthquakes started to propagate towards the south-west of the mainshock after about 20 days, outside the main aftershock area.

The distribution of early aftershocks has often been used as a proxy to infer the extent and geometry of major earthquake ruptures in a mainshock-aftershock sequence[29]. Here we show that the early increase in the area of the aftershock sequence is continuous, probably due to

the prevalence of early afterslip[30]. Using early aftershocks to define the main rupture area seems therefore inherently arbitrary, as it is completely dependent on the cutoff time. However, there is a clear, $3.5 \times 6$ km gap in aftershock seismicity downdip of the mainshock epicentre, visible for the entire postseismic period (Fig. 3c). This gap probably represents the area where most of the accumulated strain was released by the mainshock[31,32], and is therefore a conservative estimate of the rupture size, where slip most likely exceeds one-third of the maximum slip[33]. North of the seismic gap, there is a topographic high in seismicity (Fig. 3 and Extended Data Fig. 2). This topographic high could have played a role in the rupture nucleation at its leading edge, whereas the stress shadow updip could have prevented the propagation of the rupture in that direction[34]. Meanwhile, the seismicity is 'thick' at the downdip edge of the seismic gap, especially in the south where several fault planes are superposed over a thickness of more than 600 m (Fig. 2d and Extended Data Fig. 2). Here the geometrical discontinuity of the rupture fault plane could have contributed to stopping the earthquake rupture[35].

The spatial variability of the aftershock expansion is also worth exploring. The clear deceleration and acceleration that we see towards the north (Fig. 3a) around the time of the largest aftershock resembles other cases of step-like aftershock expansion thought to be related to fault segmentation[36], to pore fluid pulses[37] or to variations in frictional properties and/or the presence of subducted structure[38]. In our case, the aftershock front was stalled at what is likely to be the edge of the $M_w$ 4.8 aftershock, where earthquakes form a straight lineament (Fig. 3c), hinting at the presence of a frictional or structural barrier. This boundary occurs in a region where the interface seismicity falls on a single thin fault, whereas both to the north and south of the boundary seismicity clearly occurs on several connected faults (Fig. 3d and Extended Data Fig. 2), suggesting that fault structure influenced the expansion of aftershocks. Mapping and studying such structures in detail could be crucial for our understanding of seismic versus aseismic slip in those regions.

## Afterslip deformation processes

Considering the subduction interface as a shear zone of finite thickness with potentially several fault planes has important implications for the way that we understand and model seismic and aseismic slip. With our thickness estimates we can place bounds on the shear strain rate associated with the afterslip, given as $\gamma = \frac{v}{H}$ in which $v$ is the sliding velocity and $H$ is the thickness[3]. For this, we estimate the average afterslip displacement in two ways. We first assume that the afterslip moment is equal to 10–30% of the mainshock moment[39], and that it is reached by the end of our observation period. The total area of aftershocks after 1,227 h (51 days) is $170 \pm 19$ km², from which we subtract the estimated rupture area of 21–61 km² to obtain an afterslip area $A^{\mathrm{afterslip}}$ of 109–149 km² (ref. 28) (Fig. 3b). The average displacement of the afterslip is given as $D = \frac{0.2 \times M_0^{\mathrm{mainshock}}}{\mu A^{\mathrm{afterslip}}}$, where the mainshock moment $M_0^{\mathrm{mainshock}}$ is equal to $6.93 \times 10^{17}$ N m and the shear modulus $\mu$ is equal to 26 GPa (ref. 40). We obtain a displacement of 1.6–8.8 cm, whereas the average sliding velocity is equal to $3 \times 10^{-9}$ to $2 \times 10^{-8}$ m s⁻¹. Alternatively, we estimate the afterslip displacement by assuming that slip in the aseismic portions of the faults is equal to the average aftershock slip on the seismic portions of the fault. For this, we assume circular ruptures with an average stress drop $\Delta\sigma$ of 2.5 MPa with a log error of 0.4 for all aftershocks[41]. Then, from ref. 42, the rupture area of a single aftershock is given as $A = \pi \times \left( \frac{7}{16} \frac{M_0}{\Delta\sigma} \right)^{\frac{2}{3}}$ whereas the displacement[43] is given as $D = \frac{M_0}{\mu A}$. We can therefore calculate the average seismic displacement on the megathrust as $D_{\mathrm{average}} = \frac{\sum D_i A_i}{A_{\mathrm{total}}}$ where $D_i$ and $A_i$ are the displacement and area of aftershocks within 800 m of the megathrust and $A_{\mathrm{total}}$ is the total area where seismic ruptures occur, overlapping or not. Using this approach, the average displacement on the seismic portions of the fault is equal to $2.7 \pm 1.7$ cm, which overlaps with our first estimate

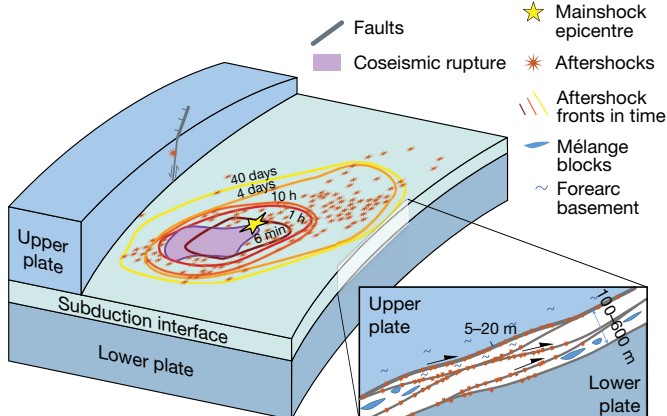

**Fig. 4 | Summary cartoon of the main features illuminated by the seismicity.** Aftershocks occur primarily around the mainshock rupture and expand along the plate interface as a function of the log of time, reflecting the likely influence of afterslip. We show that plate interface earthquakes occur on a network of metres-thick, subparallel, anastomosing faults at the plate interface, occurring over a thickness of a few hundred metres.

and yields an average sliding velocity of $6 \pm 4 \times 10^{-9}$ m s⁻¹ with a maximum sliding velocity of $3 \times 10^{-5}$ m s⁻¹ in the first 5 min after the mainshock. If the aseismic deformation is distributed throughout the whole roughly 20 m width of the fault zones we observe with the seismicity, the strain rate is about $10^{-10}–10^{-9}$ s⁻¹ on average, reaching roughly $10^{-6}$ s⁻¹ immediately after the mainshock and providing a lower bound for the strain rate. An upper bound for the strain rate is provided by assuming a fault thickness-displacement scaling relationship as classically found for brittle faults[44]. The observed afterslip displacements at $2.7 \pm 1.7$ cm would indicate a thickness of up to roughly 1 cm within the afterslip fault core[1,3,44] and the corresponding strain rate would be around $10^{-7}–10^{-6}$ s⁻¹ on average and reach up to roughly $10^{-3}$ s⁻¹ after the mainshock.

The lower bound values of $10^{-10}$ to $10^{-6}$ s⁻¹ would be consistent with solution precipitation creep at low pore fluid pressures[3], but tomographic data along the Ecuadorian subduction zone suggest that afterslip regions tend to correlate with anomalously high $v_P/v_S$ (P- and S-wave velocity, respectively) ratios, and hence probably high pore fluid pressures[45]. Although foliated argillites may contribute to high $v_P/v_S$ (ref. 46), high pore pressures at convergent plate boundaries are also indicated by low megathrust earthquake stress drops[47], the force balance at convergent margins[48] and the presence of characteristic hydrofractures in exhumed plate interface zones[3]. In a region of high pore fluid pressure, the above range of strain rates cannot be accommodated by solution precipitation creep. Depending on the true thickness of the afterslip fault core—somewhere between the full fault width of roughly 20 m and that deduced from scaling relationships, roughly 1 cm—brittle creep appears the most likely mechanism for afterslip[3].

## Megathrust structure and impact on slip

Our findings are summarized in Fig. 4. Thanks to our high precision microearthquake locations in the aftermath of a moderate-sized earthquake, we can illuminate the structure of the subduction interface at a depth of 15–20 km and examine in detail the spatio-temporal evolution of the seismicity. Looking at the timing of earthquakes, we find that fluid diffusion cannot explain the expansion of aftershocks in the region, meaning these are mostly controlled by afterslip. Meanwhile, the locations of earthquakes show that the seismic deformation is not constrained to a single continuous fault plane, but rather occurs on anastomosing, mostly subparallel metres-thick active faults that are sometimes superposed, in agreement with the geological record.

This fault zone complexity probably affects the propagation of aftershocks and afterslip in the region, thus challenging our understanding of the link between these two phenomena and the structures that host them.

Our findings show the importance of conceptualizing the megathrust as a fault network, particularly when discussing strain accumulation and seismogenesis, rather than viewing it as a single plane. This is especially relevant for creating more realistic models of earthquake rupture and aseismic slip in subduction zones, and for a better assessment of the related earthquake hazard.

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

## Methods

### Event detection and location
Detection and association of phases was carried out using the machine learning tools available as part of the SeisBench package[9]. The continuous waveforms were scanned using PhaseNet[50], obtaining more than 6 million picks. Phase association was performed using the HEX method[8], initially obtaining 112,336 events. This initial set of events was then relocated using NonLinLoc[51,52] in a local one-dimensional velocity model[10], and filtered in terms of number of phases and location accuracy, obtaining a high-quality subset of 1,775 events.

### Magnitude calculation
Local magnitudes were calculated using maximum peak-to-peak amplitudes and the original relation proposed by Bakun and Joyner[53] for California earthquakes (MCAL). These magnitudes were then benchmarked against the local magnitudes calculated by the Instituto Geofisico, Escuela Politecnica Nacional, Quito, Ecuador (MIG) for a subset of 148 common events. The obtained linear regression was MCAL = MIG + 0.49, which was then used to calculate the final local magnitude. We calculate the $b$ value and magnitude of completeness in Extended Data Fig. 6.

### Focal mechanisms
Focal mechanisms for 24 selected events were computed using the grid-search algorithm HASH[54,55], which takes the P-wave first motion polarities and S/P amplitude ratios as input data. For this, we used only stations within 50 km epicentral distance and clear P-wave onset.

### Relative relocation
We use the TomoDD software[11] with an existing 3D velocity model[45] to obtain accurate relative earthquake locations by using catalogue and correlation double-difference times. Cross-correlations are calculated on data from the vertical component filtered between 2 and 10 Hz, using a 2.1 and 3.3 s window for the P- and S-waves, respectively. If the correlation is above 0.75, the differential time is then used for relocation in TomoDD, with a weight equal to the square of the correlation coefficient. We require a minimum of eight phase links to define a pair and set the maximum separation between pairs to 6 km. Doing so still keeps all 1,571 successfully relocated events within a single cluster during the relocation. We also calculate errors through bootstrapping, because TomoDD does not give accurate error estimates. Two tests are used to determine errors. In the first jackknife test, we remove one station per event pair for 100 iterations to test the impact of the network on our results. In the second test, we add random noise to all our double-difference measurements for 100 iterations. The error is taken as the standard deviation of the locations obtained throughout the 100 iterations. The median location errors for the jacknife and bootstrap tests are 37 and 70 m, respectively, and the median errors in interevent distance are 22 and 33 m.

### Determining the main plane of seismicity
We define a plane representing the new local plate interface based on our data. Because most earthquakes between 15 and 25 km in depth with a relative location error below 75 m seem well-aligned on the plate interface, we simply fit a plane through them with a least-squares inversion. We then remove outliers, defined as earthquakes with depths within the lowest and highest 5% relative to the main plane. With this new set of data, we fit our final plane of seismicity, again with a least-squares inversion. The error is then calculated through a bootstrapping and jackknife test, in which we remove 10% of events and add a random error proportional to the event's location error to the locations for 500 iterations. Overall, our plane has a strike of 17.7 ± 0.6° and a dip of 23.3 ± 0.2°, with the seismicity forming an 872 ± 27 m thick band. However, these errors are probably underestimated due to the flexure

and topography of the plate interface, as well as to the possible inclusion of interplate seismicity in the calculation.

### Finding similar and repeating earthquakes
We perform cross-correlation on the vertical component, this time with a filter of 4–12 Hz and a window of twice the S and P time. Low correlations are discarded if the associated signal to noise ratio is below 15 to ensure that noise does not limit our classification.

Similar earthquakes are defined as earthquakes with correlations above 0.75 for at least 80% of usable stations (example in Extended Data Fig. 7). The 0.75 threshold was chosen on the basis of the histogram of cross-correlations found in Extended Data Fig. 8. Indeed, this value separates two main groups of events: non-similar events, whose correlation can be positive or negative but with a peak absolute value around 0.25, and similar events, which have high correlations with a peak around 0.95. This indicates that they probably are close in space and have similar focal mechanisms, and thus probably occur on the same or on close, parallel faults. We further define these individual fault planes with the three-point method[56], using earthquakes within the family that have location errors of less than 20 m relative to each other and less than one tenth of the interevent distance. Fisher statistics are used to find the best-fit plane and its associated uncertainties. We retain well-defined fault planes if the value of $K$ is greater than five, the angle uncertainty is under 20° and the probability is greater than 80%.

Repeating earthquakes are defined as earthquakes with correlations above 0.92 for at least 80% of usable stations. In addition, their interevent distance must be smaller or equal, within errors, to the radius of the largest event's rupture. The latter is calculated assuming that moment magnitude $M_w = M_L - 0.5$ where $M_L$ is the local magnitude, with a stress drop of 3 Mpa (ref. 41). By using both criteria, we ensure that repeaters truly rupture the same area. We find ten pairs of repeating earthquakes, one of which occurs exclusively in the preseismic period and one of which has one preseismic and one postseismic event. The preseismic repeater occurs in the crustal cluster, whereas all other repeaters occur near the plate interface. However, it is likely that this repeating earthquake catalogue is incomplete due to the small time-window examined, as many asperities probably did not have time to reload.

### Geostatistical analysis and seismicity thickness calculations
The variations in elevation of earthquakes relative to the best-fit plane are analysed using an experimental half-variogram, defined as:

$$\gamma(d) = \frac{1}{2N(d)} \sum_{i=1}^{N} [h(x_0 + d) - h(x_0)]^2$$

where $h(x_0)$ is the elevation of a given earthquake relative to the best-fit plane, and $h(x_0 + d)$ is the elevation of another earthquake at a distance $d$. $N(d)$ is the number of pairs at a distance $d$ and $\gamma(d)$ is the semivariance of the elevation difference as a function of interevent distance. The half-variogram reaches a constant value, known as the sill, at a given distance, known as the correlation length or the effective range. At that distance, separated values of elevation are no longer correlated at all, meaning this corresponds to a characteristic width of topographic features or of patches of a given thickness. Meanwhile, the sill is related to the characteristic height of topographic features, as it indicates the overall variance of the elevation. The value of the half-variogram at zero is known as the nugget, and in our case is linked to the thickness of the seismicity at a given point. We define the thickness as the width containing 95% of the seismicity, approximated as four times the square root of the nugget.

We calculated the half-variogram every 50 m over 4 km, using more than 108,000 earthquake pairs occurring within 800 m of the main plane of seismicity with a relative elevation error of less than 20 m and a distance error of less than 25 m (Extended Data Fig. 9). All errors

cited here are calculated by removing 10% of the data and adding noise before fitting either model for 100 iterations. Our data are best fit by the exponential model, which yields a correlation length of $1.82 \pm 0.03$ km with a nugget of $3{,}511$ m$^2$ and a sill of $43{,}980$ m$^2$, for which we estimate the thickness of the seismicity at around $237 \pm 18$ m and the characteristic height of topographic features at around $210 \pm 1$ m. This estimate of thickness is almost equal to the 227 m calculated using only event pairs less than 50 m apart. By comparison, with the less well-fitting spherical model, we obtain a seismicity thickness of $377 \pm 11$ m and a characteristic height and correlation length of topography of $194 \pm 1$ m and $1.59 \pm 0.04$ km, respectively.

As the thickness of the seismicity is particularly model dependent, we calculate an extra upper bound. To do so, we divide the megathrust into smaller regions of $1.8 \times 1.8$ km, equal to the exponential model's correlation length, thereby removing the effect of plate flexure and wide topographic features. We remove squares containing fewer than ten events and only keep results if the standard deviation of the jack-knife test we perform is smaller than 75 m. Doing this, we obtain an average thickness of 455 m with 95% of values falling between 164 and 958 m. This is larger than our previous estimates, as it is further affected by narrow or steep topographic features. This approach also allows us to map thickness variation and large-scale topographic features (Extended Data Fig. 2), thus giving an overall view of the structure of the plate interface.

## Data availability

Earthquake catalogues created during this study are available in a data supplement to this paper at https://doi.org/10.35097/1921, along with the absolute and differential traveltimes of all earthquakes and the location errors calculated by bootstrapping.

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

**Acknowledgements** This work was supported by the ANR FLUID2SLIP project, grant no. ANR-21-CE49-0012 of the French Agence Nationale de la Recherche. Funding from the Geophysical Institute at KIT for field work and C.C. is acknowledged. We thank everyone involved in the deployment of land stations during the HIPER experiment, including W. Acero, I. Tapa, D. García, C. Viracucha, Christian Espín, J. Santo, F. Mejía, V. Reyes-Wagner, S. León-Ríos, L. Mejía and B. Braszus.

**Author contributions** C.C. performed the relative relocation with TomoDD, the search for repeaters and most of the data analysis (assisted by the other authors of the manuscript). H.A.-D. computed initial absolute locations, magnitudes and focal mechanisms. A.R. developed the conceptual idea of this study and acquired funding. M.S., A.G. and A.R. coordinated data collection. All authors assisted in discussing the results and editing the manuscript.

**Funding** Open access funding provided by Karlsruher Institut für Technologie (KIT).

**Competing interests** The authors declare no competing interests.

**Additional information**
**Correspondence and requests for materials** should be addressed to Andreas Rietbrock.

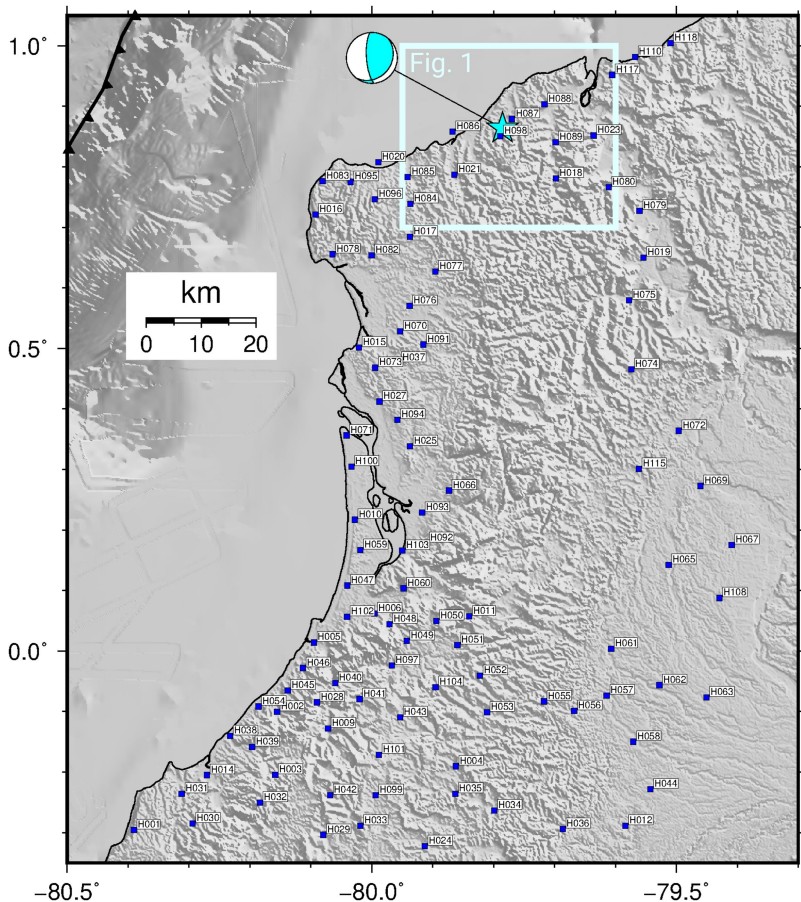

**Extended Data Fig. 1 | Map of seismic stations (blue squares).** The focal mechanism from the Global CMT catalogue is shown in light blue, at the relocated location. The light blue rectangle shows the region mapped in Fig. 1. Grey shading is used to illuminate the topography and bathymetry from the Global Earth Relief Grids dataset, available from GMT (NASA, 2013).

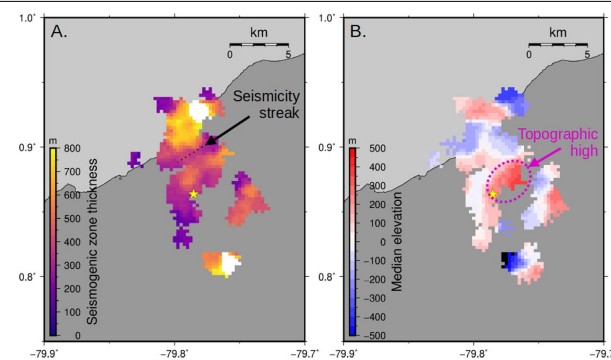

**Extended Data Fig. 2 | Maps of seismicity within 800 m of the main plane with a location error of 75 m or less.** A: Thickness containing 95% of plate interface events within a 0.9 km radius. While apparent thickness can also be affected by steep topography, intraplate earthquakes and sparse data, it can serve as a first-order proxy for fault structure. In this map, areas with thicknesses above ~450 m have multiple parallel planes of seismicity. B: Median elevation of events relative to the main seismicity plane, calculated within a 1.6 km radius.

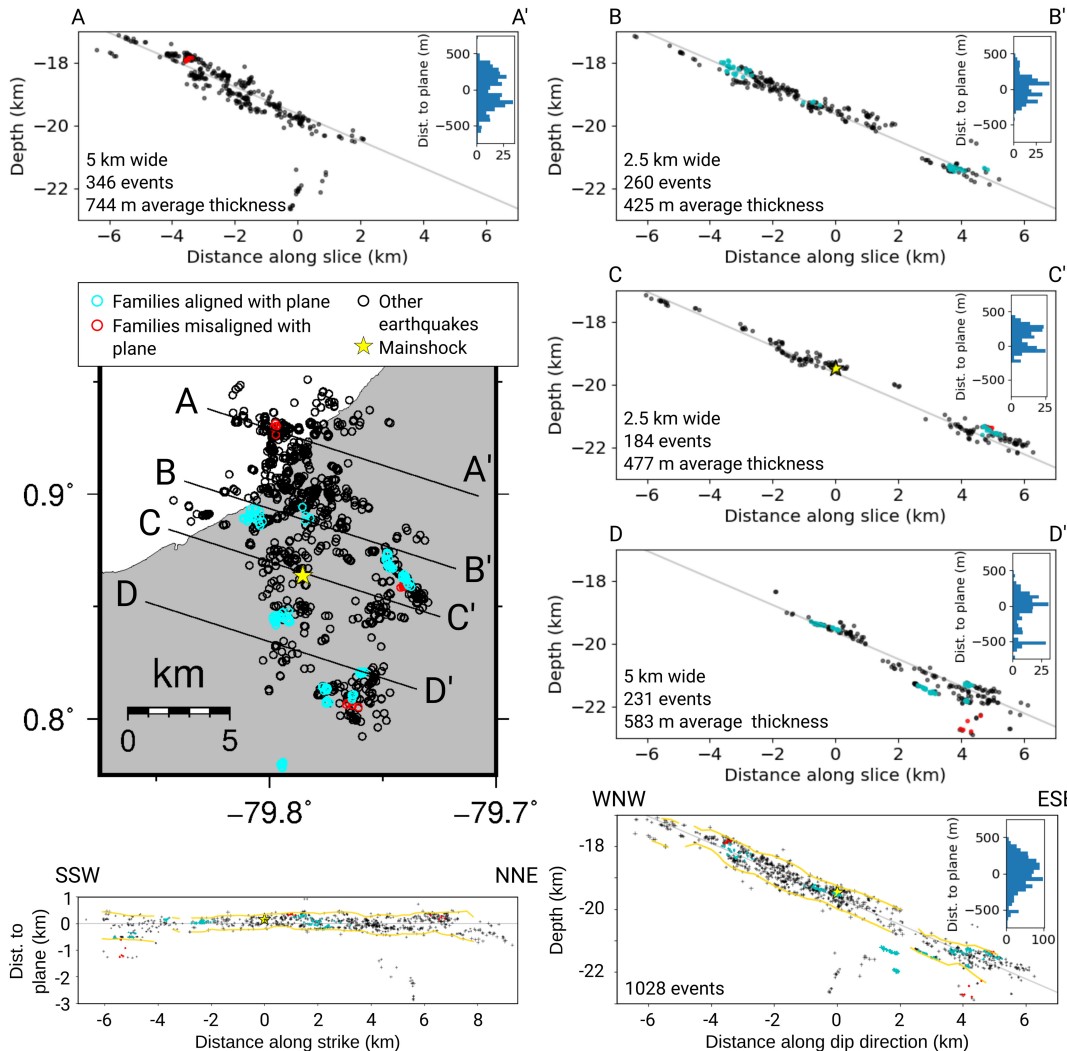

**Extended Data Fig. 3 | Additional cross-sections of earthquakes with location errors < 75 m.** Blue and red circles show families of similar earthquakes (CC > 0.75), aligned with the main plane of seismicity with an angle lower or higher than 30 degrees respectively. In the cross-sections for the whole region, 95% of earthquakes are found between the yellow lines.

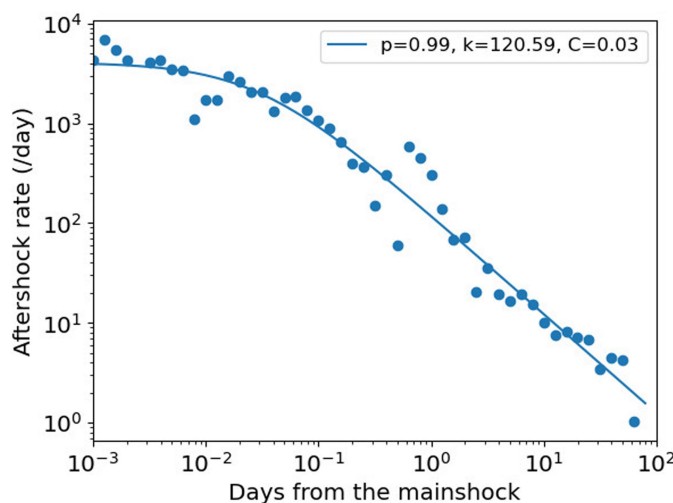

**Extended Data Fig. 4 | Omori plot for earthquakes occurring at the plate interface.** Blue dots show the real aftershock rates, while the blue line is the best-fit line with a least-squares inversion.

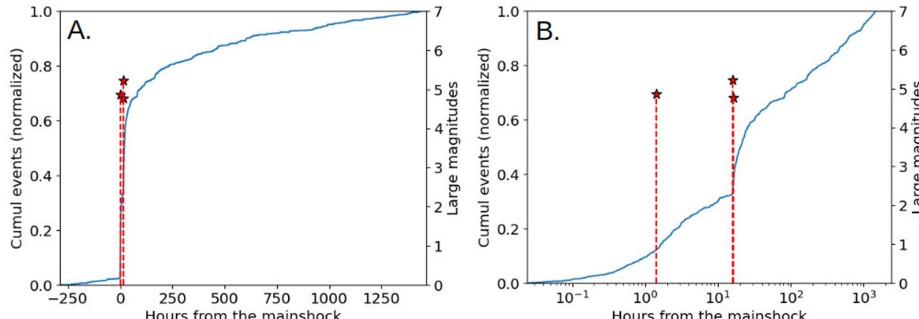

**Extended Data Fig. 5 | Cumulative number of earthquakes at the plate interface.** A: Cumulative number of earthquakes in linear time, with large magnitude aftershocks as red stars. B: Cumulative number of aftershocks in log time, with large magnitude aftershocks as red stars.

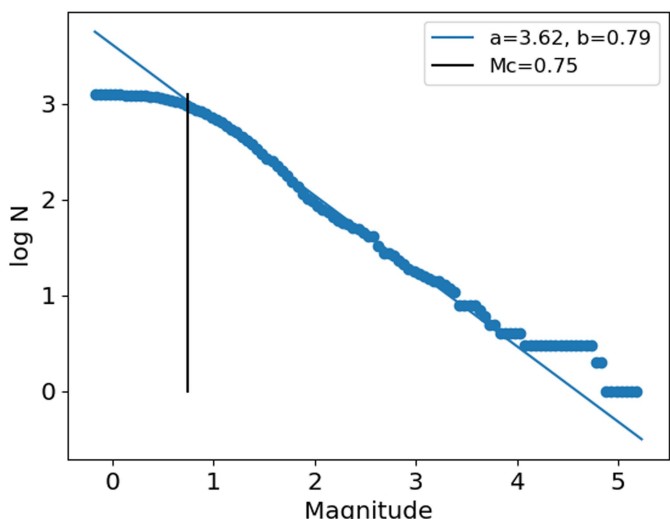

**Extended Data Fig. 6 | B-value plot for earthquakes occurring at the plate interface.** Blue dots show the real number of earthquakes above a given magnitude, while the blue line shows the best-fit line obtained with a least-squares inversion assuming a magnitude of completeness $M_c$ of 0.75.

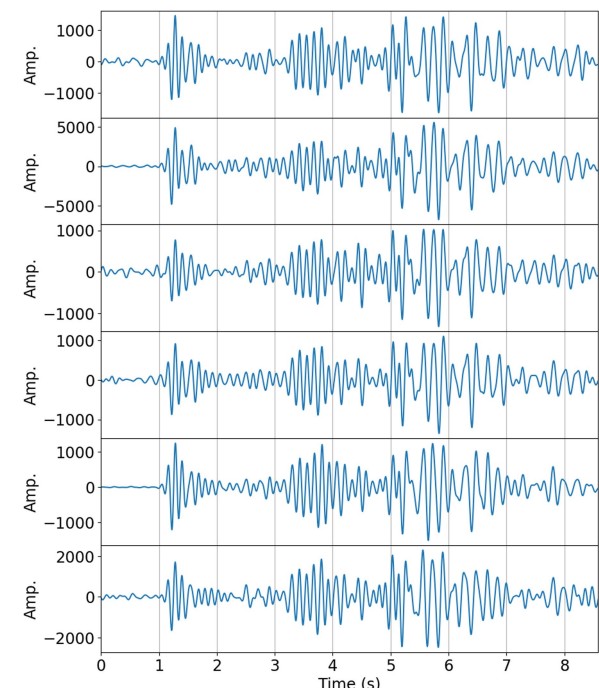

**Extended Data Fig. 7 | Family 5 of similar earthquakes, recorded at station H018.** Earthquake magnitudes are between 0.74 and 1.29. Data is filtered between 2 and 10 Hz.

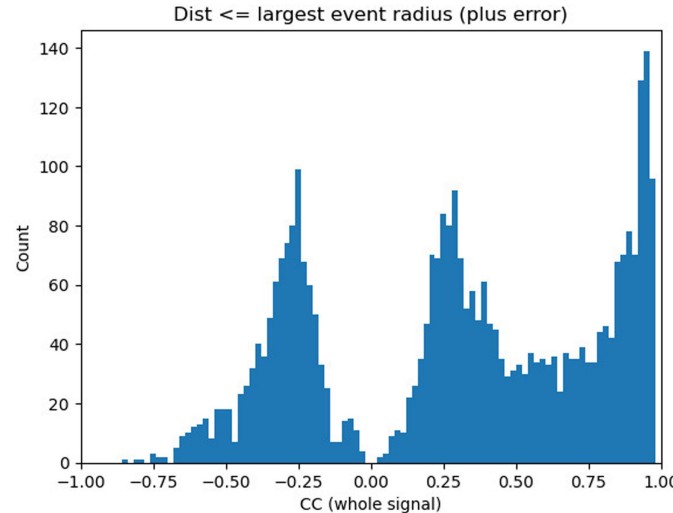

**Extended Data Fig. 8 | Histogram of cross-correlations calculated over the whole signal for all event pairs within a rupture length plus error of each other.** The rupture length is calculated assuming $M_w = M_L - 0.5$ and a stress drop of 3 Mpa.

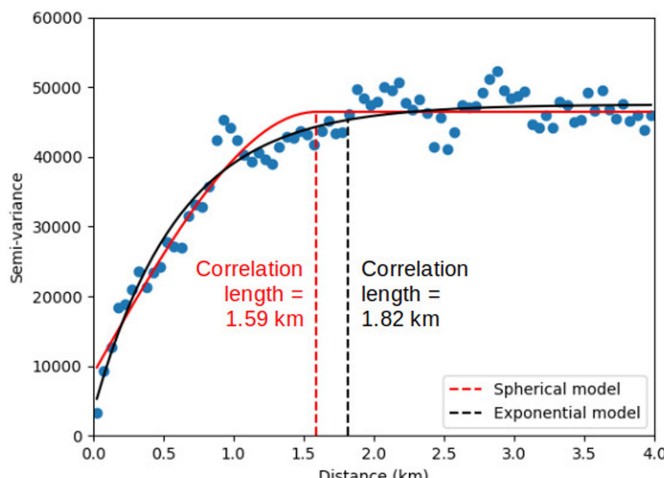

**Extended Data Fig. 9 | Semi-variogram of the earthquake elevation relative to the main plane of seismicity.** Only earthquakes located within 800 m of the main seismicity plane are considered. Additionally, earthquake pairs are used only if their relative location error is below 25 m and their relative elevation error is below 20 m. A total of 108224 pairs are used.