## [Peer Review File · Nature]

Manuscript Title: Seismological evidence for a multi-fault network at the subduction interface

Reviewer Comments & Author Rebuttals

Reviewer Reports on the Initial Version:

Referee #1:

This study uses data collected during a temporary deployment of a dense set of short-period seismometers following the Mw 7.8 2016 Ecuador earthquake. This deployment recorded a Mw 5.8 event in 2022 northeast along the megathrust relative to the 2016 rupture, and this event had a productive aftershock sequence that was well-detected and for which 1500 relative locations were determined. The precision of the relative locations is better than for many megathrust sequences because of the location below land at relatively shallow depth of 15 to 20 km (a result of underthrusting of the buoyant Carnegie Ridge). A 3D velocity structure for the regional structure for the region and double difference relative locations enabled locations purportedly with location uncertainties less than 60 m.

A relatively tight aftershock distribution with quasi-planarity is evident in cross-sections through the relocated events (Fig. 1), with some profiles showing what appear to be parallel lineations of events that are used to infer offset subparallel shear structures within a ~900 m thick band (median thickness of around 390m with 90% of values between 160 and 680 m. Space-time expansion of the aftershock sequence and a relatively aseismic region within a ring of aftershocks are used to explore the sequence, favoring afterslip driven aftershock expansion (rather than diffusionally driven). Inferences about strain rates in the plate interface shear zone are made which hinge on the assumed thickness of the distribution.

The data analysis is competently executed, and the algorithms for picking arrivals and locating events are now well-established. It is hard to judge the validity of uncertainty estimates, as this may vary with event magnitude and position, but there is no question that the geometry and instrument coverage is about as favorable as we can hope to see for subduction interfaces below land. The inferred mainshock slip zone has small dimensions, as expected from the magnitude, and this region has probably ruptured in much larger slip events (e.g., the 1906 and 1958 megathrust ruptures), so inferring overall behavior of the megathrust from a small Mw 5.8 event and its aftershocks does not clarify the relationship of this sequence to the damage zone from larger events, but the study is interesting and well done overall.

I do think the assessments of event precision are rather optimistic and the seismicity profiles would greatly benefit from inclusion of some uncertainty bounds (likely as a function of magnitude and position with respect to the array; profile 2A is offshore while 2C is under land, so azimuthal span of the stations varies). In general, I think this is an interesting study, but the overall significance is not so clear to me. I note that the recent much larger 2020 Simeonof and 2021 Chignik earthquakes along the Alaska Peninsula region have all of the coseismic slip within a zone where reflection imaging shows a wide band 2-4 km thick of reflectors. It is unclear how concentrated the slip is within the reflector band, but it may be that larger events with coseismic slip over the entire region of this sequence rupture a narrow band, distinct from the multi-bands activated in this 2022 aftershock sequence. So, I am not sure how significant the small activity is tectonically.

Referee #2:

Review for – Chalumeau et al.

Seismological evidence for a multi-fault network at the subduction interface

To the authors and editor:

Thank you for the opportunity to review this exciting new work. The authors have demonstrated the structure of the seismogenic plate boundary in unprecedented spatial detail that should be propagated into earthquake cycle models, as it may impact on our understanding of deformation mechanisms in the plate boundary, the evolution of loading on the megathrust during and after earthquakes, strain distribution and rate, and therefore estimates of stress. This study is novel, pushes the boundaries of our understanding, leverages a fantastic and unique opportunity for imaging plate boundary seismicity and appears robust in all aspects. Nature should publish this paper after some polishing as described below.

I am not equipped to critique most of the methods, particularly the specific details of the relative relocations of earthquake hypocenters which are fundamental to the work. The only methodological question I ask is to include more documentation of how the clusters of earthquakes were selected for 3-point calculations. These selections are essential for establishing the existence of multiple linked but differently orientated source structures.

Assuming the methods are confirmed, the conclusions are robust and I think actually more impactful than the authors have claimed. This being Nature, I think it would be appropriate to increase the description of potential impact and importance as represented in the summary paragraph, introduction, and conclusions. I have made some suggestions below.

I hope to see this paper accepted and I am excited to see these results.

Christie Rowe

Suggested improvements:

1. The motivation. In the abstract and introduction, the motivation for the study should be more specific to these results and more generalized for earthquake science. It is great to see that these high-resolution relocations can illuminate the plate boundary fault structure and especially satisfying that they are consistent with the exhumed rock record. However, it may not be obvious to readers why it is critical to know the detailed structure of these plate boundary faults. The motivation should also suggest how this insight can be used in detail to create more naturalistic rupture models, or to link afterslip and aftershocks, or to understand the role of geometric barriers during different parts of the earthquake cycle. If any of these uses for your results may have an impact on hazard assessment for example, this should be suggested in the end of the abstract.
2. The two-dimensional slices through the relocated earthquakes are helpful but actually seeing these structures in 3D would be more compelling and have the possibility for showing spatial relationships between fault strands more clearly. Also please help me connect together the locations of maps in Figures 1, 2, 3 with more annotations.

Minor comment throughout – edit to be consistent with verb tense. Should be present tense to describe what you see in the data and past tense to describe past events. For example, on line 58, “A secondary cluster of seismicity occurs in the upper crust” suggests it is ongoing, so it would be more correct to say “A secondary cluster of seismicity occurred in the upper crust” as the time period of this observation is in the past. Similarly the next sentence “It is active intermittently” should be “It was active”.

Line edit suggestions:

42. The text refers to Fig S3, but the sentence is describing the station locations which are shown on Fig 1.

56. Focal mechanisms show mostly oblique thrust faulting

Fig 1: many small comments:

- the cross sections A-D are not mentioned in the caption, which should say what aperture around the cross section line and duration in time is represented in the seismicity shown. Each has a straight dipping line that does not match the relocated earthquakes, is this the plate boundary from some other source like the slab model? Are the slab interface contours plotted from sea level or from land surface? What are the E labels on the cross sections?
- The figure shows mostly onshore seismometers and H086 that might be offshore, is that correct? Is this the only OBS used in the study?

59: "upper crust" -- is that subducting crust or upper plate? On cross section D there are clusters both above and below the interface.

Fig 2: A list of questions and suggestions:

- The formatting is rather awkward with a lot of white space and none of the different panels in the figure are aligned nor similarly scaled. Can you improve the aesthetics of this figure and perhaps create a more efficient use of space?
- Is it possible to relate the traces of the visual clusters in the cross sections with their orientation on the stereonet? E.g. the stereonet shows two clusters that are approximately horizontal planes and one with a west dip, it would be interesting to see the positions of those planes relative to the plate boundary-parallel planes if they are visible in one or more cross sections.
- Annotate the depth/slice sections with their orientation (e.g. A is a box with WNW-ESE and NNE-SSW sides, I assume due to the apparent dip that the profile is showing the WNW-ESE side of the A box). The orientations should be bearings (e.g. 290-110 instead of W-E) to avoid confusion of imprecise directions between text and figures.
- Please also look at the text styles, which are different on each of the separate parts of the figure and should be uniform in font and size.
- Each individual part needs to have a subfigure label so it can be referred to in the text.
- Finally, the families aligned with planes are all plotted in cyan in the map and cross sections, but black in the stereonet, while cyan is used for a best fit plane for all seismicity. It would be more intuitive if the same colors were the same group of earthquakes.
- N-S section is not actually N-S and is labeled as strike direction, but is also not parallel to the contours of interface depth from Fig 1A. Please revise to make consistent. Similar to W-E section which is not oriented W-E and also apparently not a dip section, why was this orientation chosen? It seems the cross sections in Fig 1 are also similar orientation to this line, but not exactly, perhaps it is convergence direction? If so please label.

62. Fitting a plane.... Please include in the methods a description of how you determined which groups of earthquakes to fit with a plane, was there any statistical method used or by visual inspection to find different potential planar clusters?

69. Plate interface topography amplitude varies as a function of wavelength – are these yellow line zones (and the average number of 640) at all varying with the analyzed length? It might be worth parameterizing this a little differently, especially if there is more long-wavelength curvature for the clusters that are longer in the dip direction (ranging from 1-15 km different lengths of sections in Fig 2). Because I think what you're getting at here is the total seismogenic thickness on a local scale, it might be slightly overestimated due to longer wavelength topography, flexure, lower plate normal faults, etc. If I understand the meaning. Also wondering what are the error bars on the exact depth of the reference plane, are these propagated through the thickness calculation?

73. I think it's a stretch to say that there is any seismogenic thickness "typically observed" in subduction zones – it could be reworded to note that the estimates of seismogenic thickness have thinned over the last few decades as presumably resolution has increased.

101. What are the thicknesses reported for the reflective band? Is it on par with the short-term seismic zone thickness? I think it is an open question how these reflective zones are produced at

depth but we must consider the possibility that these are foliated mica-rich zones rather than pore fluid – or both factors may contribute.

105. "... increase in thickness at depth (from ~200 m to ~4 km) ..." clarify that these estimates come from Alaska, not Ecuador, this reinforces the novelty of your work and also I think it is a nice independent confirmation that these dimensions might be meaningful.

~120-140. This section of text describes the aftershock propagation front which is displayed in various ways in Fig 3, but the text lacks figure calls to help readers align the description with the images. Please add figure calls to the text and add annotations of specifically mentioned features to the figures, e.g. the NE-SE alignment of seismicity mentioned in line 133. A time animation from an oblique 3D view would be so powerful here, is it possible to add one? Perhaps in supplementary material if Nature does not allow embedding of animations?

140: I don't understand what is meant by "... and not triggered by any large ($M > 4$) earthquake". Maybe this means to say that subsequent $>M4$ aftershocks do not trigger any apparent changes in the aftershock patterns?

148: the gap in aftershocks mentioned in the text is not that obvious to me in Fig 3 – which panel should I look? Probably best to annotate this in the figure, at least in panel A and perhaps also in C and D if the reader is meant to see the same low-density patch in all three time views.

149: small note but "accumulated stress was fully released by the mainshock" sounds to me like you are claiming the stress drop was complete across the whole mainshock rupture area, is that right? If not, just remove the word "fully" or replace it with "significantly" or something similar.

150: this sentence seems a bit circular, you suggest a minimum bound on the rupture area and then note that it is smaller than expected, but this seems fully consistent.

153-6. Figure 2 must show the topographic high you describe in either A, B or C but Fig 2 does not show the seismic gap/mainshock patch clearly so I cannot figure out which topography in cross section is being referred here? More annotation and more precise figure calls will fix this. In general if any feature is mentioned in the text there should be a big arrow on a map or cross section with the same words so reader can find it easily. I may be confused about the location of the low density aftershock patch (which is not presently labeled on any figures) because I assumed it is coincident with the earliest aftershock density contour, therefore the north nucleation and stress shadow? Is "behind" south? Doesn't make spatial sense from the figures.

169-170. Difficulty understanding the spatial association of Figures 2A and S8 here – more annotation needed to make clear how these fit together. The boundary is not obvious in any figures.

Fig. 3 – Many small comments/questions:

- maybe this is not reasonable to ask, but Fig 3 A showing the contours of aftershocks with log time is fascinating and I would love to see it in other dimensions – in cross section? In 3D.
- I am curious whether the aftershocks progression goes in parallel on different fault strands or whether they all proceed together in X-Y. The rupture area estimates in B are not so different in area to the first aftershocks contour (log hours = -1), is this a reasonable approximation of the rupture patch?
- dots for the events are very tiny, either stretch the map larger or increase the dot size for figure 3A.
- Plot 3B – I don't understand what the "large magnitudes" axis is describing, or which of the data and curves on the plot it applies to? Not mentioned in the caption.
- Maps in figs 1-2 show the coastline with grey land and white water. In Fig 3 the maps are all grey with an unlabeled wavy line from NE-SW which I think is also the coastline? Make water white

if so, that way is more consistent and intuitive to interpret.

177-210. This is a useful exploration of parameter space, I know there are length limitations but the assumed (or suggested) values are not completely justified, errors are not presented or propagated. It would be more robust and perhaps not much longer to frame this as some reasonable values just to place bounds on strain rates (and indicate high/low for reasonable parameter range).

205. – the high v_p/v_s ratios could be explained by some combination of elevated pore fluid volume (not sensitive to pore pressure, although people often convolve volume with pressure) or by the development of phyllosilicate foliations in the shear zone
useful sources:

Miller et al. <https://agupubs.onlinelibrary.wiley.com/doi/full/10.1029/2021GL094511>

Kirkpatrick et al. <https://www.nature.com/articles/s43017-021-00148-w>

208-211. In Rowe et al. 2011 and Rowe et al. 2013 (that's me) the ~ 1 cm fault slip surfaces are considered to be likely seismic slip surfaces, but the 7-31 m wide strands (which I understand to be analogous to your 0-40 m wide aftershock zones) are high strain surfaces at more like intermediate strain rates (in Rowe et al. 2011 we suggested that afterslip is fast but distributed strain in this zone). I consider it very unlikely that afterslip happens on a single discrete 1 cm surface. So the strain rate for afterslip if distributed in one of your fault strands is more like 10^{-9} – 10^{-10} /s – not very diagnostic of any particular deformation mechanism. So, if you think your 22 \pm 20 m-thick aftershock zone is actually position uncertainty but the aftershocks are all on one single plane, then leave the text as is, or if you agree with my way of thinking that the aftershocks are distributed in space within a 22 \pm 20 m wide fast creeping zone, then modify this sentence. I don't know which is right.

216-7. The fault strands seem to be anastomosing, that makes it possible to smoothly transition creep from one segment to another. As written, this text might be misunderstood as fault segments that are not connected.

Fig S3 – might be nice to outline the area of Fig 1 on here, I see now that this is much larger area so Fig 1 isn't actually showing the complete station coverage from the deployment. I wonder if there is room to add station names or a link to the complete station map if it can be downloaded somewhere. Please cite the datasets for onland topography and offshore – is that bathymetry? Describe in caption.

Fig. S8 – fascinating way of visualizing these data. In the areas where the seismogenic zone thickness is large (white patches to the north and south) are these areas where there might be multiple planes of earthquakes that vertically overlap. The text (Line 169) refers to a "boundary" between the thick and thin seismogenic zones in this figure, but I cannot see a sharp boundary between north and south, more like thin in the middle and thick to north and south edges? Please annotate the features you wish readers to see.

Author Rebuttals to Initial Comments:

Referee #1:

This study uses data collected during a temporary deployment of a dense set of short-period seismometers following the Mw 7.8 2016 Ecuador earthquake. This deployment recorded a Mw 5.8 event in 2022 northeast along the megathrust relative to the 2016 rupture, and this event had a productive aftershock sequence that was well-detected and for which 1500 relative locations were determined. The precision of the relative locations is better than for many megathrust sequences because of the location below land at relatively shallow depth of 15 to 20 km (a result of underthrusting of the buoyant Carnegie Ridge). A 3D velocity structure for the regional structure for the region and double difference relative locations enabled locations purportedly with location uncertainties less than 60 m.

A relatively tight aftershock distribution with quasi-planarity is evident in cross-sections through the relocated events (Fig. 1), with some profiles showing what appear to be parallel lineations of events that are used to infer offset subparallel shear structures within a ~900 m thick band (median thickness of around 390m with 90% of values between 160 and 680 m). Space-time expansion of the aftershock sequence and a relatively aseismic region within a ring of aftershocks are used to explore the sequence, favoring afterslip driven aftershock expansion (rather than diffusively driven). Inferences about strain rates in the plate interface shear zone are made which hinge on the assumed thickness of the distribution. The data analysis is competently executed, and the algorithms for picking arrivals and locating events are now well-established. It is hard to judge the validity of uncertainty estimates, as this may vary with event magnitude and position, but there is no question that the geometry and instrument coverage is about as favorable as we can hope to see for subduction interfaces below land. The inferred mainshock slip zone has small dimensions, as expected from the magnitude, and this region has probably ruptured in much larger slip events (e.g., the 1906 and 1958 megathrust ruptures), so inferring overall behavior of the megathrust from a small Mw 5.8 event and its aftershocks does not clarify the relationship of this sequence to the damage zone from larger events, but the study is interesting and well done overall.

Dear Referee #1,

Thank you very much for your interesting comments. We have answered them to the best of our ability, and hope you will be satisfied with our corrections. You mention that this region has probably ruptured in much larger events, and thus inferring the behavior of the megathrust from a Mw 5.8 event and its aftershocks does not clarify the relationship of this sequence to the damage zone from larger events. However, while the mainshock does appear to be located near the southern edge of the 1958 rupture and within the 1906 rupture (Figure 1), these rupture areas are too poorly constrained for us to be able to analyze the spatial relationship between our sequence and past large events. This being the case, the study of intermediate-sized earthquakes is itself important, and can help deepen our understanding of slip processes in general. In this study, we show that seismic activity is occurring on several distinct faults at the downdip edge of the 2022 mainshock rupture, as

well as along its northern edge. This may indicate that the transition from a single fault to multiple faults plays an important role in earthquake segmentation in subduction zones, which we hope will be explored further through modeling and the study of other earthquakes.

Figure 1: Modified from Chalumeau et al. (2021). Seismotectonic features in the study region. Interseismic coupling (Nocquet et al., 2014) is shown in brown color scale. SSEs are shown in pink and the 2016 Pedernales afterslip is shown in purple. White stars and white lines show the epicenters and approximate rupture areas of past megathrust earthquakes (Kanamori and McNally, 1982; Mendoza and Dewey, 1984). The yellow star and yellow line show the epicenter and the 1 m contour of the rupture zone of the 2016 Pedernales earthquake (Nocquet et al., 2017). The blue star shows the 2022 rupture.

I do think the assessments of event precision are rather optimistic and the seismicity profiles would greatly benefit from inclusion of some uncertainty bounds (likely as a function of magnitude and position with respect to the array; profile 2A is offshore while 2C is under land, so azimuthal span of the stations varies).

This is a fair concern. As TomoDD does not provide robust error estimates, we obtain them from bootstrapping and jackknifing, as is commonly done (Waldhauser and Ellsworth, 2000). In the previous version of this manuscript, we considered the location error of a given earthquake to be the median deviation from its location after 100 iterations of our

bootstrapping and jackknife test. We have now changed this to be the standard deviation of the locations, which did increase our calculated errors, but does not affect the conclusions of this paper. Wherever possible, we have added error estimates in our manuscript, including in the cross-sections of figure 2 and 3 where we now show error bars for the earthquake locations.

In general, I think this is an interesting study, but the overall significance is not so clear to me. I note that the recent much larger 2020 Simeonof and 2021 Chignik earthquakes along the Alaska Peninsula region have all of the coseismic slip within a zone where reflection imaging shows a wide band 2-4 km thick of reflectors. It is unclear how concentrated the slip is within the reflector band, but it may be that larger events with coseismic slip over the entire region of this sequence rupture a narrow band, distinct from the multi-bands activated in this 2022 aftershock sequence. So, I am not sure how significant the small activity is tectonically.

The reviewer raises an interesting point. This study allows us to visualize the distribution of slip during a Mw 5.8 earthquake and its subsequent aftershock sequence, revealing an intricate network of active faults. The mainshock appears to initiate where we observed a single fault, and since we are in a place where we cannot image the subduction interface by seismic imaging, it provides a unique view of the deformational zone within this section of the Ecuadorian subduction interface. Of course, in other regions, high resolution seismic imaging could have a better spatial coverage and provide a view of the plate interface within the rupture areas of great megathrust earthquakes (eg. Kuehn, 2019). Kuehn (2019) and Li et al. (2015) show that the thickness of the reflectors in the seismogenic zone is around ~2-4 km, and through modeling they infer that this must be caused by a single 100-250 m thick low-velocity zone. This is in agreement with our results, since our total seismic thickness is around 200-300 m. However, seismic imaging cannot give information on “active” structures within this zone, or show how they are activated on the timescale of the seismic cycle. This is where we believe our study to be truly unique, and why we believe it will provide crucial information, particularly to the modeling community. By showing evidence for multiple active fault segments within a fault network at the plate interface, we allow for the possibility of creating more realistic subduction models, which will ultimately help us determine what role fault geometry plays in the nucleation and propagation of large earthquakes. Additionally, our study sheds light on the location of aseismic slip, which plays a critical role in the seismic cycle and can influence the propagation of larger seismic ruptures.

Referee #2:

To the authors and editor:

Thank you for the opportunity to review this exciting new work. The authors have demonstrated the structure of the seismogenic plate boundary in unprecedented spatial detail that should be propagated into earthquake cycle models, as it may impact on our understanding of deformation mechanisms in the plate boundary, the evolution of loading on the megathrust during and after earthquakes, strain distribution and rate, and therefore estimates of stress. This study is novel, pushes the boundaries of our understanding, leverages a fantastic and unique opportunity for imaging plate boundary seismicity and appears robust in all aspects. Nature should publish this paper after some polishing as described below.

I am not equipped to critique most of the methods, particularly the specific details of the relative relocations of earthquake hypocenters which are fundamental to the work. The only methodological question I ask is to include more documentation of how the clusters of earthquakes were selected for 3-point calculations. These selections are essential for establishing the existence of multiple linked but differently orientated source structures.

Assuming the methods are confirmed, the conclusions are robust and I think actually more impactful than the authors have claimed. This being Nature, I think it would be appropriate to increase the description of potential impact and importance as represented in the summary paragraph, introduction, and conclusions. I have made some suggestions below.

I hope to see this paper accepted and I am excited to see these results.
Christie Rowe

Dear Christie Rowe,

Thank you for your kind and detailed comments. We have modified our manuscript accordingly, and hope that you find this new version satisfactory.

Regarding the way that earthquake clusters were selected, we have added in the methods and supplementary material the following, which we hope clarifies this point:

L 430-434: "The 0.75 threshold was chosen based on the histogram of cross-correlations found in Figure S10. Indeed, this value separates two main groups of events: non-similar events, whose correlation can be positive or negative but with a peak absolute value around 0.25, and similar events, which have high correlations with a peak around 0.95."

The positive and negative portions of the cross-correlation histogram are nearly symmetrical at low absolute values, reflecting the randomness of the correlation between non-similar events. Thus we choose a cross-correlation threshold of 0.75 because the negative portion of the histogram shows us that non-similar events cut off at that threshold. That being said, we note that we can still observe mostly subparallel, 0-40 m thick fault planes if we lower the correlation threshold to 0.7, or if we increase it to 0.8, as we demonstrate in the figure below.

Figure 2: Thickness and orientations of planes for families defined using a threshold correlation coefficient of 0.7, 0.75 and 0.8.

Suggested improvements:

1. The motivation. In the abstract and introduction, the motivation for the study should be more specific to these results and more generalized for earthquake science. It is great to see that these high-resolution relocations can illuminate the plate boundary fault structure and especially satisfying that they are consistent with the exhumed rock record. However, it may not be obvious to readers why it is critical to know the detailed structure of these plate boundary faults. The motivation should also suggest how this insight can be used in detail to create more naturalistic rupture models, or to link afterslip and aftershocks, or to understand the role of geometric barriers during different parts of the earthquake cycle. If any of these uses for your results may have an impact on hazard assessment for example, this should be suggested in the end of the abstract.

The summary paragraph has been partially rewritten and we hope it better emphasizes the motivations. We have also added this sentence to the conclusions:

L 323-325: “This is especially relevant for creating more realistic models of earthquake rupture and aseismic slip in subduction zones, and for a better assessment of the related earthquake hazard.”

2. The two-dimensional slices through the relocated earthquakes are helpful but actually seeing these structures in 3D would be more compelling and have the possibility for showing spatial relationships between fault strands more clearly. Also please help me connect together the locations of maps in Figures 1, 2, 3 with more annotations.

Thank you for your suggestions on how to improve figures throughout the manuscript. Annotations have indeed been added to the figures to make them more clear. Unfortunately, we were not able to make static 3D plots that adequately conveyed the geometry of the seismic sequence.

Minor comment throughout – edit to be consistent with verb tense. Should be present tense to describe what you see in the data and past tense to describe past events. For example, on line 58, “A secondary cluster of seismicity occurs in the upper crust” suggests it is ongoing, so it would be more correct to say “A secondary cluster of seismicity occurred in the upper crust” as the time period of this observation is in the past. Similarly the next sentence “It is active intermittently” should be “It was active”.

We have addressed this comment throughout the manuscript and hope it is now entirely consistent.

Line edit suggestions:

42. The text refers to Fig S3, but the sentence is describing the station locations which are shown on Fig 1.

This has been rectified. Both of these figures are now cited, as they both show the locations of stations (L 40).

56. Focal mechanisms show mostly oblique thrust faulting

Indeed, this has been corrected in the text (L 66).

Fig 1: many small comments:

- the cross sections A-D are not mentioned in the caption, which should say what aperture around the cross section line and duration in time is represented in the seismicity shown. Each has a straight dipping line that does not match the relocated earthquakes, is this the plate boundary from some other source like the slab model? Are the slab interface contours plotted from sea level or from land surface? What are the E labels on the cross sections?

These comments have been taken into account and the figure and caption changed accordingly:

L 51-57: “The locations for these focal mechanisms are also shown as labels in the cross-section. The Font et al. (2013) slab model is plotted as a black line in each cross-section and as dashed contours in map view. Nearby seismic stations are shown as yellow squares. The inset shows the convergence velocity between the Nazca plate and the

North Andean Sliver. Cross-sections A-D each have a width of 5 km and a bearing of 108 degrees, corresponding to the dip direction of the main plane of seismicity."

- The figure shows mostly onshore seismometers and H086 that might be offshore, is that correct? Is this the only OBS used in the study?

This is actually a land station, as no OBS were used. The coastline in the previous version of this figure was off by as much as a few km in some places, and this has now been corrected by taking the coastline from the Global Earth Relief Grids dataset available from GMT.

59: "upper crust" -- is that subducting crust or upper plate? On cross section D there are clusters both above and below the interface.

This was indeed the upper plate, as we have now corrected (L 70).

Fig 2: A list of questions and suggestions:

- The formatting is rather awkward with a lot of white space and none of the different panels in the figure are aligned nor similarly scaled. Can you improve the aesthetics of this figure and perhaps create a more efficient use of space?
- Is it possible to relate the traces of the visual clusters in the cross sections with their orientation on the stereonet? E.g. the stereonet shows two clusters that are approximately horizontal planes and one with a west dip, it would be interesting to see the positions of those planes relative to the plate boundary-parallel planes if they are visible in one or more cross sections.
- Annotate the depth/slice sections with their orientation (e.g. A is a box with WNW-ESE and NNE-SSW sides, I assume due to the apparent dip that the profile is showing the WNW-ESE side of the A box). The orientations should be bearings (e.g. 290-110 instead of W-E) to avoid confusion of imprecise directions between text and figures.
- Please also look at the text styles, which are different on each of the separate parts of the figure and should be uniform in font and size.
- Each individual part needs to have a subfigure label so it can be referred to in the text.
- Finally, the families aligned with planes are all plotted in cyan in the map and cross sections, but black in the stereonet, while cyan is used for a best fit plane for all seismicity. It would be more intuitive if the same colors were the same group of earthquakes.
- N-S section is not actually N-S and is labeled as strike direction, but is also not parallel to the contours of interface depth from Fig 1A. Please revise to make consistent. Similar to W-E section which is not oriented W-E and also apparently not a dip section, why was this orientation chosen? It seems the cross sections in Fig 1 are also similar orientation to this line, but not exactly, perhaps it is convergence direction? If so please label.

These comments have been taken into account and the figure and caption changed accordingly (L 131-152).

62. Fitting a plane.... Please include in the methods a description of how you determined which groups of earthquakes to fit with a plane, was there any statistical method used or by visual inspection to find different potential planar clusters?

The relevant section has been added to the methods, detailing the fitting procedure and giving an estimate of the errors:

L 409-422: “Determining the main plane of seismicity:

We define a plane representing the new local plate interface based on our data. Since most earthquakes between 15 and 25 km depth with a relative location error under 75 m seem well-aligned on the plate interface, we simply fit a plane through them with a least-squares inversion. We then remove outliers, defined as earthquakes with depths within the lowest and highest 5% relative to the main plane. With this new set of data, we fit our final plane of seismicity, again with a least-squares inversion. The error is then calculated through a bootstrapping and jackknife test, in which we remove 10% of events and add a random error proportional to the event’s location error to the locations for 500 iterations. Overall, our plane has a strike of 17.7 ± 0.6 degrees and a dip of 23.3 ± 0.2 degrees, with the seismicity forming an 872 ± 27 m thick band. However, these errors are likely underestimated due to the flexure and topography of the plate interface, as well as to the possible inclusion of interplate seismicity in the calculation.”

69. Plate interface topography amplitude varies as a function of wavelength – are these yellow line zones (and the average number of 640) at all varying with the analyzed length? It might be worth parameterizing this a little differently, especially if there is more long-wavelength curvature for the clusters that are longer in the dip direction (ranging from 1-15 km different lengths of sections in Fig 2). Because I think what you’re getting at here is the total seismogenic thickness on a local scale, it might be slightly overestimated due to longer wavelength topography, flexure, lower plate normal faults, etc. If I understand the meaning. Also wondering what are the error bars on the exact depth of the reference plane, are these propagated through the thickness calculation?

We agree with the reviewer, that when we calculate the topography and thickness in a square of a given size, they do vary with the size of the region considered. This is why we have expanded the section on the calculation of thickness and topography in the methods, under the header “Geostatistical analysis and seismicity thickness calculations”, and hope to have made the limitations in our analysis more clear.

L 469-495: “We define the thickness as the width containing 95% of the seismicity, approximated as 4 times the square root of the nugget.

We calculated the half-variogram every 50 m over 4 km, using over 108000 earthquake pairs occurring within 800 m of the main plane of seismicity with a relative elevation error of less than 20 m and a distance error of less than 25 m (Figure S9). All errors cited here are calculated by removing 10% of the data and adding noise before fitting either model for 100 iterations. Our data is best fit by the exponential model, which yields a correlation length of 1.82 ± 0.03 km with a nugget of 3511 m² and a sill of 43980 m², for which we estimate the thickness of the seismicity at around 237 ± 18 m and the characteristic height of topographic features at around 210 ± 1 m. This estimate of thickness is almost equal to the 227 m calculated using only event pairs less than 50 m apart. By comparison, with the less well-fitting spherical model, we obtain a seismicity thickness of 377 ± 11 m and a characteristic height and correlation length of topography of 194 ± 1 m and 1.59 ± 0.04 km respectively.

Since the thickness of the seismicity is particularly model-dependent, we calculate an additional upper bound. To do so, we divide the megathrust into smaller regions of 1.8×1.8 km, equal to the exponential model’s correlation length, thereby removing the effect of plate

flexure and wide topographic features. We remove squares containing less than 10 events and only keep results if the standard deviation of the jackknife test we perform is smaller than 75 m. Doing this, we obtain an average thickness of 455 m with 95 % of values falling between 164 and 958 m. This is larger than our previous estimates, as it is additionally affected by narrow or steep topographic features. This approach also allows us to map thickness variation and large-scale topographic features (Figure S2), thus giving an overall view of the structure of the plate interface.”

73. I think it's a stretch to say that there is any seismogenic thickness “typically observed” in subduction zones – it could be reworded to note that the estimates of seismogenic thickness have thinned over the last few decades as presumably resolution has increased.

This was reworded accordingly (L 79-81: “This is smaller than what has been previously observed seismologically in subduction zones (Nippres and Rietbrock, 2007), likely due to our better resolution.”).

101. What are the thicknesses reported for the reflective band? Is it on par with the short-term seismic zone thickness? I think it is an open question how these reflective zones are produced at depth but we must consider the possibility that these are foliated mica-rich zones rather than pore fluid – or both factors may contribute.

The reflective band in the seismogenic zone is under 2 km thick, and is modeled by Li et al. (2015) as a 100-250 m thick low velocity zone, which is similar to our seismic thickness. This point has been highlighted in the text.

L 120-123: “in Alaska, seismic imaging shows a marked increase in thickness associated with the change from a singular localized shear zone in the seismogenic zone (100-250 m thick LVZ) to a wide deformation zone downdip (~4 km thick zone with multiple LVZs).”

105. “... increase in thickness at depth (from ~200 m to ~4 km) ...” clarify that these estimates come from Alaska, not Ecuador, this reinforces the novelty of your work and also I think it is a nice independent confirmation that these dimensions might be meaningful.

This has been clarified in the text (see above).

~120-140. This section of text describes the aftershock propagation front which is displayed in various ways in Fig 3, but the text lacks figure calls to help readers align the description with the images. Please add figure calls to the text and add annotations of specifically mentioned features to the figures, e.g. the NE-SE alignment of seismicity mentioned in line 133. A time animation from an oblique 3D view would be so powerful here, is it possible to add one? Perhaps in supplementary material if Nature does not allow embedding of animations?

We have modified Figure 3 and added figure calls in the text, which should make this section more clear. Movies of the cross-sections are provided in the extended data to make the time evolution of the sequence more clear.

140: I don't understand what is meant by "... and not triggered by any large ($M > 4$) earthquake". Maybe this means to say that subsequent $>M4$ aftershocks do not trigger any apparent changes in the aftershock patterns?

We meant that no large aftershocks occurred after 20 days, when this new migration of seismicity to the south occurs. We have changed this in the text.

L 181-183: "Subsequently, small ($M < 4$) earthquakes started to propagate towards the south-west of the mainshock after about 20 days, outside of the main aftershock area."

148: the gap in aftershocks mentioned in the text is not that obvious to me in Fig 3 – which panel should I look? Probably best to annotate this in the figure, at least in panel A and perhaps also in C and D if the reader is meant to see the same low-density patch in all three time views.

The annotation was added in figure 3.

149: small note but "accumulated stress was fully released by the mainshock" sounds to me like you are claiming the stress drop was complete across the whole mainshock rupture area, is that right? If not, just remove the word "fully" or replace it with "significantly" or something similar.

This sentence was reworded in the text (L 193-194).

150: this sentence seems a bit circular, you suggest a minimum bound on the rupture area and then note that it is smaller than expected, but this seems fully consistent.

This has been reworded to make the sentence less redundant (L 194-196).

153-6. Figure 2 must show the topographic high you describe in either A, B or C but Fig 2 does not show the seismic gap/mainshock patch clearly so I cannot figure out which topography in cross section is being referred here? More annotation and more precise figure calls will fix this. In general if any feature is mentioned in the text there should be a big arrow on a map or cross section with the same words so reader can find it easily. I may be confused about the location of the low density aftershock patch (which is not presently labeled on any figures) because I assumed it is coincident with the earliest aftershock density contour, therefore the north nucleation and stress shadow? Is "behind" south? Doesn't make spatial sense from the figures.

We have added several figure calls (L 158, 176, 177, 206, 213, 216) and expanded, annotated and reorganized figure 3 to ensure the clarity of this section. In particular, the topographic high and the seismic gap are both highlighted in Figure 3.

169-170. Difficulty understanding the spatial association of Figures 2A and S8 here – more annotation needed to make clear how these fit together. The boundary is not obvious in any figures.

The location of the boundary is now indicated in Figure 3, as well as in Figure S8.

Fig. 3 – Many small comments/questions:

- maybe this is not reasonable to ask, but Fig 3 A showing the contours of aftershocks with log time is fascinating and I would love to see it in other dimensions – in cross section? In 3D.

- I am curious whether the aftershocks progression goes in parallel on different fault strands or whether they all proceed together in X-Y. The rupture area estimates in B are not so different in area to the first aftershocks contour (log hours = -1), is this a reasonable approximation of the rupture patch?

This is a very interesting point, although a difficult one to answer given the sparsity of earthquake locations in some areas of the plate interface. Sadly, we do not know of any possibility to make available a timelapse of a 3D plot with the paper. We do however provide time animations of the Figure 3 cross-section, which is parallel to the main direction of propagation, showing how different fault strands are activated after the mainshock and the large aftershock. From the timelapse after the large aftershock especially, we see that different fault strands often host earthquakes at around the same time. However, during the initial propagation of the aftershocks, some fault strands arguably seem to become active before others. Unfortunately, without a larger and denser dataset, we cannot confidently conclude that propagation speed is truly different on the different fault strands that make up the plate interface.

- dots for the events are very tiny, either stretch the map larger or increase the dot size for figure 3A.

- Plot 3B – I don't understand what the "large magnitudes" axis is describing, or which of the data and curves on the plot it applies to? Not mentioned in the caption.

- Maps in figs 1-2 show the coastline with grey land and white water. In Fig 3 the maps are all grey with an unlabeled wavy line from NE-SW which I think is also the coastline? Make water white if so, that way is more consistent and intuitive to interpret.

These modifications have indeed been made to the figure and caption. To keep a good contrast between the time contours and the background, we elected to make the sea a lighter gray rather than white, but we hope the figure has been made more readable nevertheless.

177-210. This is a useful exploration of parameter space, I know there are length limitations but the assumed (or suggested) values are not completely justified, errors are not presented or propagated. It would be more robust and perhaps not much longer to frame this as some reasonable values just to place bounds on strain rates (and indicate high/low for reasonable parameter range).

We agree with the reviewer and have added error bounds to all our estimates in this section (L 261-281). Please note however that a few of our assumptions have an impact on the final number that is hard to evaluate. The first estimate of afterslip velocity relies on the assumption that the afterslip lasts until the end of the study period, and no further. The second method assumes that the seismic displacement from aftershocks at the plate interface is equal to the aseismic displacement happening around them. This relies on two assumptions. The first is that no aseismic slip occurs on a seismic asperity during the interseismic time, which may not be true (Chen and Lapusta, 2009). The second assumption

is that no aseismic deformation is occurring above or below the fault plane on which the asperity is located, which once again may not be true everywhere. As such, the number we derive with this method is likely itself a lower bound.

205. – the high vp/vs ratios could be explained by some combination of elevated pore fluid volume (not sensitive to pore pressure, although people often convolve volume with pressure) or by the development of phyllosilicate foliations in the shear zone
useful sources:

Miller et al. <https://agupubs.onlinelibrary.wiley.com/doi/full/10.1029/2021GL094511>

Kirkpatrick et al. <https://www.nature.com/articles/s43017-021-00148-w>

Indeed, the study by Miller is very interesting. However, there are several aspects that are at odds with assigning high Vp/Vs ratios to foliated argillites in subduction channel material only. The first is the fact that stress drops of megathrust earthquake events that have broken the entire interface typically are in the range of 5-20 Megapascals (e.g. Wang et al., 2019), requiring very low effective coefficients of friction (c. 0.032). At the depths of 15-25 km this requires very significant fluid overpressures. Second, force balance studies of convergent margins (e.g. Dielforder, Hetzel and Oncken, 2020) find similar very low effective coefficients of friction limiting maximum stresses to about 20 MPa and therefore requiring pore fluid pressure ranges at convergent plate boundaries to be on average above 0.9. Third, the ubiquitous presence of hydrofractures mainly in and near the main faults of exhumed plate interface zones but also in their shale matrix filled in many cases with crystals exhibiting crystal faces indicates that mineral growth occurred in open fluid-filled cavities (Nüchter and Stöckhert, 2007; Oncken, Angiboust and Dresen, 2021). Keeping them open during growth requires lithostatic pore pressures for at least some time periods during evolution of the subduction channel.

Hence, because of all these observations, we think very low effective stresses at - possibly fluctuating - pore pressures from close to lithostatic to well below lithostatic are inescapable. The properties of foliated argillites as reported by Miller et al. are not in conflict with this interpretation but may add to the observed high Vp/Vs ratio as the reviewer mentions. L 295-300: "While foliated argillites may contribute to high vp/vs (Miller et al., 2021), high pore pressures at convergent plate boundaries are also indicated by low megathrust earthquake stress drops (Wang et al., 2019), the force balance at convergent margins (Dielforder et al., 2020), and presence of characteristic hydrofractures in exhumed plate interface zones (Oncken et al., 2021)."

208-211. In Rowe et al. 2011 and Rowe et al. 2013 (that's me) the ~1 cm fault slip surfaces are considered to be likely seismic slip surfaces, but the 7-31 m wide strands (which I understand to be analogous to your 0-40 m wide aftershock zones) are high strain surfaces at more like intermediate strain rates (in Rowe et al. 2011 we suggested that afterslip is fast but distributed strain in this zone). I consider it very unlikely that afterslip happens on a single discrete 1 cm surface. So the strain rate for afterslip if distributed in one of your fault strands is more like 10^{-9} – 10^{-10} /s – not very diagnostic of any particular deformation mechanism. So, if you think your 22+/-20 m-thick aftershock zone is actually position uncertainty but the aftershocks are all on one single plane, then leave the text as is, or if you agree with my way of thinking that the aftershocks are distributed in space within a 22 +/- 20 m wide fast creeping zone, then modify this sentence. I don't know which is right.

We fully agree with the reviewer that seismic slip is constrained to these thin surfaces and that the wide strands found in nature are the result of multiple seismic events and their afterslip. We do not know exactly whether afterslip of the analyzed event embraced the complete fault strand width. Assuming that the entire width of such a fault strand accumulates the afterslip of a single event however yields a lower bound on the real strain rate. If the classical fault displacement-thickness scaling relationship is assumed then the strain rates given in Oncken et al. (2021; 10^{-2} – 10^{-4} /s), might be considered an upper bound. Reality is likely to be in between these two bounds.

Finally, we agree with the reviewer's suggestion that the aftershocks are probably distributed on individual thin slip surfaces enclosed within the 21 +/- 20 m wide fault strands. This is also what is seen in exhumed field records: Multiple thin (c. 1 cm wide) principal slip surfaces that are ultracataclasite-lined are distributed within the 20-40m wide cataclasite-filled larger fault strands.

L 300-305: "In a region of high pore fluid pressure, the above range of strain rates cannot be accommodated by solution precipitation creep. Depending on the true thickness of the afterslip fault core – somewhere between the full fault width of approx. 20m and that deduced from scaling relationships, approx. 1cm, – brittle creep appears the most likely mechanism for afterslip (Oncken et al., 2021)."

216-7. The fault strands seem to be anastomosing, that makes it possible to smoothly transition creep from one segment to another. As written, this text might be misunderstood as fault segments that are not connected.

This has been rectified in the text (L 315).

Fig S3 – might be nice to outline the area of Fig 1 on here, I see now that this is much larger area so Fig 1 isn't actually showing the complete station coverage from the deployment. I wonder if there is room to add station names or a link to the complete station map if it can be downloaded somewhere. Please cite the datasets for onland topography and offshore – is that bathymetry? Describe in caption.

This has been changed in the Figure and caption.

Fig. S8 – fascinating way of visualizing these data. In the areas where the seismogenic zone thickness is large (white patches to the north and south) are these areas where there might be multiple planes of earthquakes that vertically overlap. The text (Line 169) refers to a "boundary" between the thick and thin seismogenic zones in this figure, but I cannot see a sharp boundary between north and south, more like thin in the middle and thick to north and south edges? Please annotate the features you wish readers to see.

Indeed this was not clear in the text. The "boundary" we talked about referred to this thinning between two zones where the seismicity is thicker due to the presence of multiple parallel faults. We have rephrased this section, and annotated the figure for clarity.

L 211-216: "In our case, the aftershock front was stalled at what is likely to be the edge of the Mw 4.8 aftershock, where earthquakes form a straight lineament (Figure 3C), hinting at the presence of a frictional or structural barrier. This boundary occurs in a region where the interface seismicity falls on a single thin fault, while both to the north and south of the boundary seismicity clearly occurs on multiple connected faults (Figure 3D, S2)"

References:

- Chalumeau, C. *et al.* (2021) 'Repeating Earthquakes at the Edge of the Afterslip of the 2016 Ecuadorian Mw7.8 Pedernales Earthquake', *Journal of Geophysical Research: Solid Earth*, 126(5), p. e2021JB021746. Available at: <https://doi.org/10.1029/2021JB021746>.
- Dielforder, A., Hetzel, R. and Oncken, O. (2020) 'Megathrust shear force controls mountain height at convergent plate margins', *Nature*, 582(7811), pp. 225–229. Available at: <https://doi.org/10.1038/s41586-020-2340-7>.
- Kanamori, H. and McNally, K.C. (1982) 'Variable rupture mode of the subduction zone along the Ecuador-Colombia coast', *Bulletin of the Seismological Society of America*, 72(4), pp. 1241–1253. Available at: <https://doi.org/10.1785/BSSA0720041241>.
- Kuehn, H. (2019) 'Along-trench segmentation and down-dip limit of the seismogenic zone at the eastern Alaska-Aleutian subduction zone'. Available at: <https://DalSpace.library.dal.ca//handle/10222/75145> (Accessed: 1 December 2023).
- Li, J. *et al.* (2015) 'Downdip variations in seismic reflection character: Implications for fault structure and seismogenic behavior in the Alaska subduction zone', *Journal of Geophysical Research: Solid Earth*, 120(11), pp. 7883–7904. Available at: <https://doi.org/10.1002/2015JB012338>.
- Mendoza, C. and Dewey, J.W. (1984) 'Seismicity associated with the great Colombia-Ecuador earthquakes of 1942, 1958, and 1979: Implications for barrier models of earthquake rupture', *Bulletin of the Seismological Society of America*, 74(2), pp. 577–593. Available at: <https://doi.org/10.1785/BSSA0740020577>.
- Nocquet, J.-M. *et al.* (2014) 'Motion of continental slivers and creeping subduction in the northern Andes', *Nature Geoscience*, 7(4), pp. 287–291. Available at: <https://doi.org/10.1038/ngeo2099>.
- Nocquet, J.-M. *et al.* (2017) 'Supercycle at the Ecuadorian subduction zone revealed after the 2016 Pedernales earthquake', *Nature Geoscience*, 10(2), pp. 145–149. Available at: <https://doi.org/10.1038/ngeo2864>.
- Nüchter, J.-A. and Stöckhert, B. (2007) 'Vein quartz microfabrics indicating progressive evolution of fractures into cavities during postseismic creep in the middle crust', *Journal of Structural Geology*, 29(9), pp. 1445–1462. Available at: <https://doi.org/10.1016/j.jsg.2007.07.011>.
- Oncken, O., Angiboust, S. and Dresen, G. (2021) 'Slow slip in subduction zones: Reconciling deformation fabrics with instrumental observations and laboratory results', *Geosphere*, 18(1), pp. 104–129. Available at: <https://doi.org/10.1130/GES02382.1>.
- Waldhauser, F. and Ellsworth, W.L. (2000) 'A Double-Difference Earthquake Location Algorithm: Method and Application to the Northern Hayward Fault, California', *Bulletin of the Seismological Society of America*, 90(6), pp. 1353–1368. Available at: <https://doi.org/10.1785/0120000006>.
- Wang, K. *et al.* (2019) 'Stable Forearc Stressed by a Weak Megathrust: Mechanical and Geodynamic Implications of Stress Changes Caused by the M = 9 Tohoku-Oki Earthquake', *Journal of Geophysical Research: Solid Earth*, 124(6), pp. 6179–6194. Available at: <https://doi.org/10.1029/2018JB017043>.

Reviewer Reports on the First Revision:

Referee #1:

The revision has clarified most issues raised in the reviewer comments and some figures are improved. The text is now clear and quite concise. The study is relatively high-resolution and exploits an unusually favorable geometry and station distribution for analyzing a shallow segment of megathrust. The evidence for a mix of single and multiple active quasi-parallel subfaults within the aftershock sequence for the small M5.8 mainshock is well presented. Interpreting the relocated hypocenters involves many assumptions, but the authors lay out the procedure clearly. It remains possible that variable faulted structure of a damage zone is involved, as only a small number of focal mechanisms were determined, but we have to await corresponding analysis of a larger mainshock to address some of my concerns. The references are fine. I think the paper will be of broad interest and impact, so I support publication in Nature.